# Comparison of Mid-Term Outcomes between Microhook ab Interno Trabeculotomy and Goniotomy with the Kahook Dual Blade

**DOI:** 10.3390/jcm12020558

**Published:** 2023-01-10

**Authors:** Naoki Okada, Kazuyuki Hirooka, Hiromitsu Onoe, Hideaki Okumichi, Yoshiaki Kiuchi

**Affiliations:** 1Department of Ophthalmology and Visual Science, Hiroshima University, Hiroshima 734-8551, Japan; 2Department of Ophthalmology, Kagawa University Faculty of Medicine, Kagawa District 761-0793, Japan

**Keywords:** Kahook Dual Blade, micrhhook, intraocular pressure

## Abstract

This study retrospectively examined the mid-term surgical outcomes between microhook ab interno trabeculotomy (μLOT) and goniotomy when one was using the Kahook Dual Blade (KDB) in combination with phacoemulsification in primary open-angle glaucoma and exfoliation glaucoma patients. Between December 2016 and December 2020, the current study examined 47 μLOT and 52 KDB eyes that underwent surgery. When there was a < 20% reduction in the preoperative intraocular pressure (IOP) or when the IOP was > 18 mmHg (criterion A), the IOP was > 14 mmHg (criterion B) at two consecutive follow-up visits, or when there was a requirement for reoperation, these were all considered to indicate that the surgery failed. A genetic algorithm that used the preoperative IOP was used to determine the score matching. After score matching, a total of 27 eyes were evaluated. In the μLOT and KDB groups, the mean postoperative follow-up periods were 31.2 ± 13.3 and 37.2 ± 16.3 months, respectively. The results for both of the groups show there were significant postoperative reductions in the IOP (*p* < 0.05) and medication scores (*p* < 0.05) at 6, 12, 24 and 36 months. At 12, 24, and 36 months, the probabilities of success in the μLOT and KDB groups for criterion A were 70.4% and 48.2%, 61.9% and 48.2%, and 55.0% and 48.2% (*p* = 0.32; log-rank test), respectively. For criterion B, the results for the two groups were 55.6% and 33.3%, 44.4% and 33.3%, and 44.4% and 33.3% (*p* = 0.15; log-rank test), respectively. Similar postoperative complications were found between the groups. Comparable mid-term surgical outcomes were found for the uses of μLOT and KDB.

## 1. Introduction

Glaucoma is a progressive disease that is found worldwide, and it is one of leading causes of blindness [1,2]. Currently, the only method that can definitively decrease the speed of the progression of the visual field damage is to lower the intraocular pressure (IOP) [3,4]. The recent development and improvement of new surgical technologies, such as minimally invasive glaucoma surgeries (MIGS), have helped to expand the types of procedures that can be undertaken during the surgical treatment of glaucoma [5]. MIGS are commonly defined as surgical procedures that use an ab interno approach, with the MIGS devices being divided into trabecular, suprachoroidal and subconjunctival-based groups [6]. The Kahook Dual Blade (KDB; New World Medical, Rancho Cucamonga, CA, USA) or the Tanito microhook (Inami & Co., Ltd., Tokyo, Japan) are trabecular-based devices that work by improving the trabecular outflow through Schlemm’s canal. The main sites of resistance to aqueous outflow have been shown to be the trabecular meshwork and the inner walls of Schlemm’s canal [7,8].

Ab interno trabeculotomy and cataract surgery are commonly performed together. In our previous study, we compared the surgical outcomes at 12 months between microhook ab interno trabeculotomy and goniotomy with the KDB when they were performed in combination with phacoemulsification [9]. However, there are no published studies that have made mid-term direct comparisons. Therefore, we evaluated patients who underwent phacoemulsification in primary open-angle glaucoma (POAG) and exfoliation glaucoma combined with microhook ab interno trabeculotomy (μLOT) and goniotomy with KDB, and then we compared the mid-term surgical outcomes between the two groups.

## 2. Materials and Methods

### 2.1. Patient Selection

This retrospective study, which was conducted at Hiroshima University Hospital, Japan, or Kagawa University Hospital, Japan, between December 2016 and December 2020, examined eyes that underwent phacoemulsification cataract extraction along with an intraocular lens placement that was combined with either μLOT (μLOT-Phaco) or goniotomy with KDB (KDB-Phaco). The study protocol was approved by the Institutional Review Board of the Hiroshima University or Kagawa University Faculty of Medicine. All of the subjects provided written informed consent, in accordance with the principles outlined in the Declaration of Helsinki. Furthermore, all of the patients provided standard consent to the surgery prior to the surgical procedure.

To be included in the study, all of the patients had to be ≥20 years of age and be diagnosed with POAG and exfoliation glaucoma. If the patients were diagnosed with other types of glaucoma or had a history of ophthalmic surgery, they were excluded from the study. In addition, all of the patients had to have been observed postoperatively for at least 12 months in order to be included in the study. If a patient was found to have undergone bilateral surgery, only the data from the first eye were included in the analysis.

### 2.2. Surgical Technique

During the surgical procedure, an initial 2.8 mm incision was made at the temporal cornea. Subsequently, using the standard technique, phacoemulsification was then performed. The Signature Pro (Johnson & Johnson, New Brunswick, NJ, USA) was the cataract instrument that was used, and the PCB00V (Johnson & Johnson) or XY1 (HOYA, Tokyo, Japan) intraocular lens was then implanted into the capsular bag. In order to increase the visuality of Schlemm’s canal after the cataract surgery, sufficient sodium hyaluronate (Healon^®^, Johnson & Johnson Vision, Irvine, CA, USA) was added to the anterior chamber. Subsequently, the patient’s head and microscope were tilted, which was followed by the placement of the Hill lens (Ocular Inst. Bellevue, WA, USA) on the cornea. The location of Schlemm’s canal was then identified, which was followed by the insertion of the microhook or KDB through the main incision, with a nasal and inferior site incision then being made between 120° and 180°. Each of the surgeons determined the degree of the incision and type of antibacterial and anti-inflammatory eye drops that were prescribed in accordance with their personal preferences. IOP-lowering medications were stopped at the time of the surgery, and they were resumed according to the surgeon’s discretion at the postoperative follow-up visits.

### 2.3. Outcome Measures

Success was defined in the study using a Kaplan–Meier survival analysis with the two criteria. An IOP reduction of at least 20% and an IOP of 18 mmHg or less were defined as criterion A, while an IOP reduction of at least 20% and an IOP of 14 mmHg or less were defined as criterion B. An IOP reduction was defined as (preoperative IOP—postoperative IOP)/preoperative IOP. The point at which the additional glaucoma surgery was required was defined as the failure. IOPs that corresponded to criteria A and B for up to 3 months after the surgery were not considered to be a surgical failure due to the occurrence of postoperative IOP fluctuations after the trabeculotomy [7].

### 2.4. Statistical Analysis

JMP software version 16 (SAS Inc., Cary, NC, USA) was used for all of the statistical analyses. The clinical backgrounds of the subjects were analyzed using a *t*-test for continuous variables and a chi-square test for categorical variables. Differences in the frequency of the postoperative complications between the two groups were analyzed using a chi-square test. In order to evaluate the differences within and between the two groups, every 6 months, we analyzed the mean IOP, the mean number of IOP-lowering medications, and the rate of IOP reduction. The number of eye drops was counted as two when the IOP-lowering medication was administered in a fixed-dose combination. The Wilcoxon signed-ranks test was used to evaluate the difference in the number of medications between the preoperative and postoperative values. The last observation carry-forward method was used for any missing clinical data. For the covariate, the preoperative IOP was used and matched by the propensity score (the random seed value was set to 111, and the caliper coefficient was set to 0.2). All of the data are presented as the mean ± standard deviation. Statistical significance was defined as a *p* value < 0.05.

## 3. Results

There were 27 eyes in the µLOT-Phaco group that were matched to 27 eyes in the KDB-Phaco group. No significant differences were noted between the µLOT-Phaco and KDB-Phaco groups in terms of age, gender, glaucoma type, preoperative IOP, and the number of preoperative IOP-lowering medications (Table 1). In the µLOT-Phaco group, the mean follow-up period was 31.2 ± 13.3 months, while in the KDB-Phaco group, it was 37.2 ± 16.3 months.

As compared to the preoperative levels, there was a significant decrease in the IOP in both of the groups (Table 2). The preoperative IOP was 21.7 ± 7.4 mmHg, with postoperative values of 13.2 ± 2.4 mmHg, 13.4 ± 3.3 mmHg, 12.9 ± 2.0 mmHg, and 13.8 ± 3.0 mmHg at 6, 12, 24, and 36 months, respectively. The preoperative IOP level for the KDB-Phaco group was 21.7 ± 7.3 mmHg, with postoperative levels of 14.9 ± 4.5 mmHg, 14.5 ± 5.2 mmHg, 13.7 ± 3.1 mmHg, and 15.5 ± 5.2 mmHg at 6, 12, 24, and 36 months, respectively. For all of the patient visits, significant differences were observed between the preoperative and postoperative IOPs in the µLOT-Phaco and KDB-Phaco groups. In the µLOT-Phaco group, the postoperative reductions of the IOP were 32.7 ± 23.4%, 34.7 ± 24.9%, 40.9 ± 20.2%, and 34.3 ± 26.2% at 6, 12, 24, and 36 months. The postoperative reductions found for the KDB-Phaco group at the same times were 25.9 ± 27.7%, 27.3 ± 32.2%, 31.7 ± 23.8%, and 26.4 ± 34.2%. For these two groups, no significant differences were observed (*p* = 0.10, 0.36, 0.42, 0.32).

As seen in Table 3, there was a significant decrease in both of the groups’ values after the surgery for the number of IOP-lowering medications. The number of preoperative medications in the µLOT-Phaco group was 2.9 ± 1.4, while they were 1.1 ± 1.3, 1.3 ± 1.4, 1.7 ± 1.6, and 2.3 ± 1.4 at 6, 12, 24, and 36 months postoperatively, respectively. The number of medications in the KDB-Phaco group was 3.4 ± 1.0, while they were 0.7 ± 1.2, 1.0 ± 1.2, 1.5 ± 1.3, and 2.1 ± 1.3 at 6, 12, 24, and 36 months postoperatively, respectively.

Criteria A and B for the Kaplan–Meier survival curves for the µLOT-Phaco and KDB-Phaco groups are shown in Figure 1. The survival rates for criterion A in the µLOT-Phaco group were 70.4%, 61.9%, and 55.0% at 12, 24, and 36 months postoperatively, while it was 48.2% in the KDB-Phaco group for all of the time points. There were no significant differences that we observed between the two groups. The survival rates for criterion B in the µLOT-Phaco group were 55.6%, 44.4%, and 44.4% at 12, 24, and 36 months postoperatively, while in the KDB-Phaco group, it was 33.3% for all of the time points. There were no significant observed differences between the groups.

As seen in Table 4, transient IOP elevation and hyphema were determined to be postoperative complications. The postoperative elevation of an IOP to 30 mmHg or more within 2 months was defined as a transient IOP elevation. The presence of the niveau formation was used to define the hyphema. The hyphema disappeared within 2 weeks in all of the cases. In the µLOT-Phaco group, hyphema was observed in three eyes (18.5%), while two eyes (14.8%) had hyphema in the KDB-Phaco group. There were no significant observed differences between the two groups (*p* = 0.72). In the µLOT-Phaco group, two eyes (11.1%) exhibited a transient IOP elevation, while two eyes (22.2%) in the KDB-Phaco group also exhibited a transient IOP elevation. There were no significant observed differences between the two groups (*p* = 0.27). Additional surgery was required in three eyes in the KDB-Phaco group, with one surgery for exfoliation glaucoma and the other two surgeries for POAG. The reasons for the additional glaucoma surgery did not exhibit an adequate IOP of more than 21 mmHg. There were no other additional surgeries that were required in the µLOT-Phaco group.

## 4. Discussion

There is increasing number of MIGS that are been performed, with many reports being published on the effectiveness of the procedure [10,11,12,13,14,15,16]. In our current study, we examined the long-term ab interno trabeculotomy outcomes for microhook and goniotomy when one is using KDB in combination with phacoemulsification. There have been many reports that have been published on the use of the procedure on Japanese patients, as the method of trabeculotomy using microhook was developed in Japan [10,11,16]. Although in our recent study we found that there were significant reductions in the postoperative IOP and in the number of IOP-lowering medications in both the µLOT-Phaco and KDB-Phaco groups, at 12 months postoperatively, we did not find any significant differences between the two groups in terms of the survival rates [9]. Other reports on the lack of a significant difference between the µLOT-Phaco and KDB-Phaco groups have also been published elsewhere [17]. However, as these techniques have yet to be established or widely used over a long period of time, there have been no studies that have compared the long-term outcomes of these two groups. Iwasaki et al. [18] examined POAG and exfoliation glaucoma patients and reported on the long-term results of KDB-Phaco. The postoperative survival rates at 36 months when one was using the same definitions as those defined for our study were almost identical to our results, with 52.5% for criterion A and 36.9% for criterion B. In the current study, the postoperative survival rates at 36 months were 48.2% and 33.3% for criteria A and B, respectively. Thus, even though there have yet to be any longer follow-up reports than those in this previous report for KDB, we consider the results of our study to be justified. Tanito et al. [16] have additionally reported on mid-term results of a µLOT study. Although their study also targeted Japanese patients, the follow-up period was 405 ± 327 days, which is longer than those of any of the other reports. Furthermore, they evaluated a larger number of subjects, with 560 eyes being examined. At 12 and 24 months postoperatively, the authors reported 44.6% and 32.1% for criterion A, respectively, which are lower than the values that were found in our study (70.4% and 61.9% at 12 and 24 months, respectively). However, it is difficult to truly compare these results with our current findings, as their study was likely influenced by their comparison of the two groups, µLOT alone and µLOT-Phaco.

IOP-lowering medications were stopped at the time of the surgery and resumed according to the surgeon’s discretion at the time of the postoperative follow-up visits. In the current study, the number of medications gradually increased for up to 36 months in both the µLOT-Phaco and KDB-Phaco groups. Although there was usually a decrease related to the surgical effects, it was necessary to ensure that the IOP was kept at an adequate value. Therefore, the number of IOP-lowering medications was at its lowest at 6 months, and it gradually increased thereafter.

In our current study, a single surgeon (K.H.) performed all of the goniotomies with the KDB. During these surgeries, the Schlemm’s canal incision in the KDB-Phaco group was always at 120°. However, there was interoperative variability that ranged between 120° and 180° for the µLOT-Phaco group. In our previous study that examined the survival and complication rates of hyphema and transient high IOP during the 24 month postoperative follow-up period, we found that there was no significant difference between the 120° and 180° Schlemm’s canal incisions, in addition, we found a decrease in both the IOP and in the number of IOP-lowering medications in both of the groups [19]. Thus, within the µLOT-Phaco group, the extent of the incision due to differences between the surgeons appears not to have had any influence on the results.

When we were evaluating potential complications between the two groups, our findings indicate that there were no differences in the hyphema with regard to the niveau formation and transient high IOP > 30 mmHg. Moreover, the comparisons between the two groups in the other previous studies also indicated that there were no significant differences [9,17]. During a mean follow-up period of 31.3 ± 14.8 months, Iwasaki et al. [18] reported finding that 4.1% of the overall cases required additional glaucoma surgeries. However, it should be noted that significantly more cases in a KDB-Phaco group were reported in another paper to have required reoperations in the exfoliation glaucoma group [18]. Unfortunately, in our current study, it was impossible to assess the risk factors as the number of subjects was reduced due to us using propensity scores. Additional surgeries in the KDB-Phaco group tended to be more common in the late phase, with three eyes (11.1%) requiring postoperative surgery at 15, 39, and 51 months, respectively. However, as there was no significant difference for the IOP reduction rate after the surgery, additional surgeries might very well be required in the µLOT-Phaco group if a longer follow-up period was used.

This study had several limitations. First, this was a retrospective study that did not specifically collect the data that were to be analyzed, but rather, we used data that were available from a routine clinical practice. Moreover, since the cases were matched with the propensity scores, this reduced the number of cases available, and thus, the studies with larger numbers of cases could potentially find variable results. In addition, due to the relatively small number of additional surgeries and complications, it was not possible to definitively explore all of the potential risk factors. In order to address these limitations, a further prospective, multicenter, and randomized clinical trial would need to be undertaken. Secondly, there were some missing clinical data. For example, in the Schlemm’s canal incision site, postoperative peripheral anterior synechia were seen, and thus, this could have affected the surgical outcomes due to an increased resistance to the aqueous outflow.

In conclusions, comparable mid-term surgical outcomes were observed when we were using μLOT and KDB. Moreover, when these were combined with phacoemulsification, significant reductions in the IOP were observed, and there was a reduction in the number of glaucoma medications that were required during the long-term follow-ups in open-angle glaucoma patients.

## Figures and Tables

**Figure 1 jcm-12-00558-f001:**
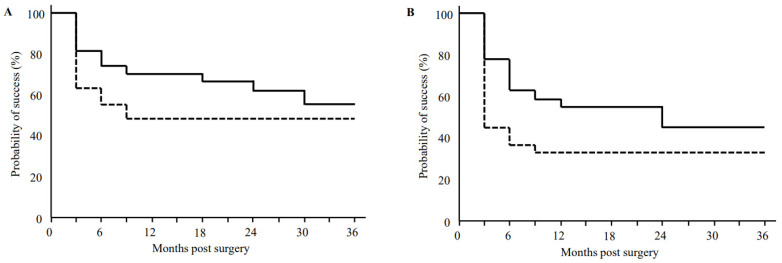
Kaplan–Meier survival curve analysis success rate of IOP control after μLOT and goniotomy with the KDB in combination with phacoemulsification according to which criterion was used to define failure. Criterion (**A**): 20% reduction in preoperative IOP value, IOP > 18 mmHg, or additional glaucoma surgery. Criterion (**B**): 20% reduction in preoperative IOP value, IOP > 14 mmHg, or additional glaucoma surgery. Solid line: μLOT-Phaco group, Dashed line: KDB-Phaco group. The numbers of patients considered to be successful for criteria A (success/total) were 20/25, 19/24, 15/23, 12/16, 9/12, and 8/12 at 6M, 12M, 18M, 24M, 30M, and 36M, respectively, in the μLOT-Phaco group and 15/27, 13/27, 11/23, 7/19, 6/17, and 5/14 at 6M, 12M, 18M, 24M, 30M, and 36M, respectively, in the KDB-Phaco group. For criteria B, the numbers considered to be successful (success/total) were 17/25, 15/24, 11/23, 6/16, 4/12, and 3/12 at 6M, 12M, 18M, 24M, 30M, and 36M, respectively, in the μLOT-Phaco group and 10/27, 9/27, 8/23, 6/19, 6/17, and 5/14 at 6M, 12M, 18M, 24M, 30M, and 36M, respectively, in the KDB-Phaco group.

**Table 1 jcm-12-00558-t001:** Clinical characteristics.

	μLOT-Phaco	KDB-Phaco	*p* Value
No. patients	27	27	
Age (years)	71.6 ± 10.5	76.0 ± 8.9	0.10
Gender (M/F)	13/14	12/15	0.78
Type of glaucoma			0.40
POAG	15	18	
Exfoliation glaucoma	12	9	
Preoperative IOP (mmHg)	21.7 ± 7.4	22.7 ± 7.3	0.97
No. IOP-lowering medication	2.9 ± 1.4	3.4 ± 1.0	0.15
Axial length (mm)	25.2 ± 2.2	25.0 ± 1.5	0.61

M: male. F: female. POAG: primary open-angle glaucoma. IOP: intraocular pressure.

**Table 2 jcm-12-00558-t002:** Differences in preoperative and postoperative IOP.

		μLOT-Phaco			KDB-Phaco		
	IOP (mmHg)	Change from Baseline (%)	*p* Value *	IOP (mmHg)	Change from Baseline (%)	*p* Value *	*p* Value **
Baseline	21.7 ± 7.4 (n = 27)			21.7 ± 7.3 (n = 27)			0.97
Month 6	13.2 ± 2.4 (n = 27)	32.7 ± 23.4	<0.01	14.9 ± 4.5 (n = 27)	25.9 ± 27.7	<0.01	0.10
Month 12	13.4 ± 3.3 (n = 27)	34.7 ± 24.9	<0.01	14.5 ± 5.2 (n = 27)	27.3 ± 32.2	<0.01	0.36
Month 18	13.3 ± 3.8 (n = 23)	36.7 ± 26.6	<0.01	14.2 ± 3.4 (n = 23)	29.8 ± 25.5	<0.01	0.38
Month 24	12.9 ± 2.0 (n-16)	40.9 ± 20.2	<0.01	13.7 ± 3.1 (n = 19)	31.7 ± 23.8	<0.01	0.42
Month 30	12.6 ± 4.4 (n = 12)	41.9 ± 29.7	<0.01	13.6 ± 3.6 (n = 17)	32.1 ± 22.8	<0.01	0.48
Month 36	13.8 ± 3.0 (n = 12)	34.3 ± 26.2	<0.01	15.5 ± 5.2 (n = 14)	26.4 ± 34.2	0.02	0.32

IOP; intraocular pressure; * Calculated using paired *t*-test for IOP between preoperative and postoperative value; ** Calculated using Student’s *t*-test for % changes from baseline between the groups.

**Table 3 jcm-12-00558-t003:** Changes in the preoperative and postoperative number of IOP-lowering medications.

	**μLOT-Phaco**		**KDB-Phaco**		
	**Number of Medications**	***p* Value ***	**Number of Medications**	***p* Value ***	***p* Value ****
Baseline	2.9 ± 1.4		3.4 ± 1.0		0.23
Month 6	1.1 ± 1.3	<0.01	0.7 ± 1.2	<0.01	0.14
Month 12	1.3 ± 1.4	<0.01	1.0 ± 1.2	<0.01	0.38
Month 18	1.3 ± 1.4	<0.01	1.2 ± 1.3	<0.01	0.78
Month 24	1.7 ± 1.6	<0.01	1.5 ± 1.3	<0.01	0.73
Month 30	2.3 ± 1.4	<0.01	1.5 ± 1.3	<0.01	0.46
Month 36	2.3 ± 1.4	0.02	2.1 ± 1.3	<0.01	0.75

IOP: intraocular pressure; * Calculated using Wilcoxon signed-ranks test for number of medications between preoperative and postoperative values; ** Calculated using Mann–Whitney’s U test for postoperative values between the groups.

**Table 4 jcm-12-00558-t004:** Postoperative complications.

	μLOT-Phaco	KDB-Phaco	*p* Value
Hyphema with niveau	3 (18.5%)	2 (14.8%)	0.72
Transient IOP elevation ≥ 30 mmHg	2 (11.1%)	2 (2 (22.2%)	0.27

IOP: intraocular pressure.

## Data Availability

The data analyzed in this study are available from the corresponding author on reasonable request.

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
