# Peer review of "Comparison of Mid-Term Outcomes between Microhook ab Interno Trabeculotomy and Goniotomy with the Kahook Dual Blade"

_jcm, 2023, doi:10.3390/jcm12020558_

Round 1

Reviewer 1 Report

1>Please make sure one tables is complete in one page, and some notes in the table shouldn't be the same size as the main text.

2>In the Discussion, if you refer to a "published report", please cite the relevant article.

Author Response

1. Please make sure one tables is complete in one page, and some notes in the table shouldn't be the same size as the main text.

We have changed. One table is in one page and the same size as the main text.

2. In the Discussion, if you refer to a "published report", please cite the relevant article.

Page 8 lines 20-26 These sentences are same reference. Our description was not adequate. We have changed the description.

Reviewer 2 Report

The manuscript (#jcm-2100106, by Okada N, et al.) is a retrospective study conducted to compare the clinical outcomes of two types of minimally invasive glaucoma surgery (MIGS) combined with cataract surgery: microhook ab interno trabeculotomy and Kahook dual blade goniotomy. The topic itself is intriguing, and the authors would have enough data to achieve their aim. However, awkward data presentation and paper writing obscure the true value of this paper. Accordingly, I recommend this manuscript for publication if the authors adequately respond to my comments below.

1) Professional English editing is mandatory as many grammatical errors are seen throughout the manuscript.

2) In Title, the word “long-term” here sounds like an exaggeration because some of the patients had been followed up for less than two years.

3) In Abstract, please amend the descriptions so that the authors can understand that this study compare two types of MIGS combined with cataract surgery.

4) On line 5 of Abstract, for "less than 20% decrease", what time point was it compared to?

5) On lines 9-10 of Abstract, “the mean follow-up periods” should be replaced by “the mean postoperative follow-up period.”

6) On line 12 of Abstract, please specify the postoperative time points that “each visit” indicates.

me

7) In Abstract, it would be nice to include the information about safety if word limit permits.

8) On lines 1 of Introduction, “The” should be replaced by “A” because there are many progressive diseases other than glaucoma.

9) On line 4 of Introduction, new “surgical” technologies?

10) On line 9 of Introduction, if the authors used Tanito microhook, please make it clear.

11) On line 24 of Introduction, “and” should be replaced by “or.”

12) In the first paragraph of Materials and Methods, although this is a retrospective study, informed consent was obtained from all participants for being include in this study. Is this true?

13) On line 21 of Materials and Methods, please add the information about product name of OVD.

14) In Materials and Methods, outcome measures should be independently indicated.

15) On line 35 of Materials and Methods, what does rate of IOP reduction mean? Please define it.

15) On line 41 of Materials and Methods, the meaning of the sentence after “if” is incomplete.

16) In Materials and Methods, the details of propensity score matching should be described.

17) On line 12 of Results, please clarify what differences were significant.

18) On line 12 of Results, please indicate the two groups.

19) On line 19 of Results, please make it clear whether preoperative medications were limited to IOP-lowering eye drops.

20) On lines 40-42 of Results, the information of the timing of additional surgeries is needed.

21) In Discussion, change in the number of glaucoma eye drops should be also discussed.

22) In Discussion, the outcomes values in previous studies should be shown and compare them with those in the present study.

23) On line 9 of 5th paragraph of Discussion, it should be shown in Materials and Methods how the authors handled “missing clinical data” in this study.

24) In Table 1, the information on LOCS III value of preoperative cataract and axial length should be provided.

25) In Table 3, “Number of medications P value” in μLOT-Phaco group is marked as bold, Is there any reason?

26) In Table 3, it would be better to use “changes” instead of “differences” because the authors do not show values of differences.

27) In Table 3, as the authors used Wilcoxon signed-ranks test here, the description Materials and Methods should be modified.

28) In Figure 1, as postoperative follow-up time differs among the patients (in other words, not all of the patients were followed up for 36 months postoperatively), the total number and success number should be provided at each time point.

29) In Table 4, please add the information about what time range was used for the assessment of postoperative complications.

30) In Table 4, please define transient IOP elevation.

Author Response

  • Professional English editing is mandatory as many grammatical errors are seen throughout the manuscript.

Our manuscript was checked by native speaker of English.

  • In Title, the word “long-term” here sounds like an exaggeration because some of the patients had been followed up for less than two years.

We have changed “long-term” to “mid-term”.

  • In Abstract, please amend the descriptions so that the authors can understand that this study compare two types of MIGS combined with cataract surgery.

Page 2 lines 1-4 We have changed.

  • On line 5 of Abstract, for "less than 20% decrease", what time point was it compared to?

Page 2 lines 5-8 IOP was compared preoperative with two consecutive follow-up visits.

  • On lines 9-10 of Abstract, “the mean follow-up periods” should be replaced by “the mean postoperative follow-up period.”

Page 2 line 11 We have changed “the mean follow-up periods” to “the mean postoperative follow-up period”.

  • On line 12 of Abstract, please specify the postoperative time points that “each visit” indicates.

Page 2 lines13-14 We have added “6, 12, 24 and 36 months postoperatively”.

  • In Abstract, it would be nice to include the information about safety if word limit permits.

Page 2 lines 18-19 We have added the information about safety.

  • On lines 1 of Introduction, “The” should be replaced by “A” because there are many progressive diseases other than glaucoma.

Page 3 line 2 We have changed “The” to “A”.

  • On line 4 of Introduction, new “surgical” technologies?

Page 3 line 5 We have added “surgical”.

  • On line 9 of Introduction, if the authors used Tanito microhook, please make it clear.

Page 3 line 10 We have added “Tanito”.

  • On line 24 of Introduction, “and” should be replaced by “or.”

Maximum line is 21 in introduction. We could not find where it was.

  • In the first paragraph of Materials and Methods, although this is a retrospective study, informed consent was obtained from all participants for being include in this study. Is this true?

We are sorry. That was our mistake. Page 4 lines 7-8 We have changed to “all patients provided standard consent for surgery prior to the surgical procedure”.

  • On line 21 of Materials and Methods, please add the information about product name of OVD.

Page 4 lines 22-23 We have added the information about product name of OVD.

  • In Materials and Methods, outcome measures should be independently indicated.

Page 5 We have indicated independently outcome measures.

  • On line 35 of Materials and Methods, what does rate of IOP reduction mean? Please define it.

Page 5 line 10-11 We have added the definition of the rate of IOP reduction.

  • On line 41 of Materials and Methods, the meaning of the sentence after “if” is incomplete.

Page 5 lines 12-14 We have changed.

  • In Materials and Methods, the details of propensity score matching should be described.

Page 6 line 1-2 We have added the details of propensity score matching.

  • On line 12 of Results, please clarify what differences were significant.

Page 6 line 19 We have changed ”two groups” to “preoperative and postoperative IOP in the µLOT-Phaco and KDB-Phaco groups”.

  • On line 12 of Results, please indicate the two groups.

Page 6 line 19 We have changed ”two groups” to “preoperative and postoperative IOP in the µLOT-Phaco and KDB-Phaco groups”.

  • On line 19 of Results, please make it clear whether preoperative medications were limited to IOP-lowering eye drops.

IOP-lowering medications were stopped at the time of the surgery and resumed according to the surgeon’s discretion at the postoperative follow-up visits. Page 5 lines 3-4 We have added this sentence.

  • On lines 40-42 of Results, the information of the timing of additional surgeries is needed.

Additional surgery was performed when IOP was more than 21 mmHg. Page 7 lines 22-23 We have added the information of the timing of the additional surgeries.

  • In Discussion, change in the number of glaucoma eye drops should be also discussed.

Page 9 lines 2-8 We have discussed about the number of IOP-lowering medications.

  • In Discussion, the outcomes values in previous studies should be shown and compare them with those in the present study.

Page 8 lines 14-18, lines 23-26 We have added the outcomes values in previous studies and compared.

  • On line 9 of 5thparagraph of Discussion, it should be shown in Materials and Methods how the authors handled “missing clinical data” in this study.

Page 5 lines 26-27 The last observation carry-forward method was used for any missing clinical data.

  • In Table 1, the information on LOCS III value of preoperative cataract and axial length should be provided.

We did not record the information on LOCS III values of preoperative cataract.

The data of axial length was added in Table 1.

  • In Table 3, “Number of medications P value” in μLOT-Phaco group is marked as bold, Is there any reason?

It was not bold in submitted Table. We do not know why it was changed.

  • In Table 3, it would be better to use “changes” instead of “differences” because the authors do not show values of differences.

We have changed “differences” to “change” in Table 3.

  • In Table 3, as the authors used Wilcoxon signed-ranks test here, the description Materials and Methods should be modified.

Page 5 lines 24-26 We have added “The Wilcoxon signed-ranks test was used to evaluate the difference in the number of medications between preoperative and postoperative values”.

  • In Figure 1, as postoperative follow-up time differs among the patients (in other words, not all of the patients were followed up for 36 months postoperatively), the total number and success number should be provided at each time point.

We have added the total number and success number in figure legend.

  • In Table 4, please add the information about what time range was used for the assessment of postoperative complications.

Page 7 lines 13-15 We have added the information.

  • In Table 4, please define transient IOP elevation.

Page 7 lines 13-14 We have added the definition of transient IOP elevation.

Round 2

Reviewer 2 Report

The manuscript was well revised, and now I recommend if for publication. Good work!